# Embodied Scene Cloning: Solving Generalization in Robotic Manipulation via Visual-Prompt Image Editing

## Abstract

Recent advancements in robotic learning have enabled robots to perform a wide range of tasks. However, generalizing policies from training environments to deployment environments remains a major challenge, and improving these policies by collecting and annotating demonstrations in target environments is both costly and time-consuming. To address this issue, we propose Embodied Scene Cloning, a novel visual-prompt-based framework that generates visual-aligned trajectories from existing data by leveraging visual cues from the specific deployment environment. This approach minimizes the impact of environmental discrepancies on policy performance. Unlike traditional embodied augmentation methods that rely on text prompts, we propose to "clone" source demonstrations into the target environment and edit it with visual prompt to effectively improve the generalization ability on specific embodied scene. Experimental results demonstrate that samples generated by Embodied Scene Cloning significantly enhance the generalization ability of policies in the target deployment environments, representing a meaningful advancement in embodied data augmentation.

## 1 Introduction

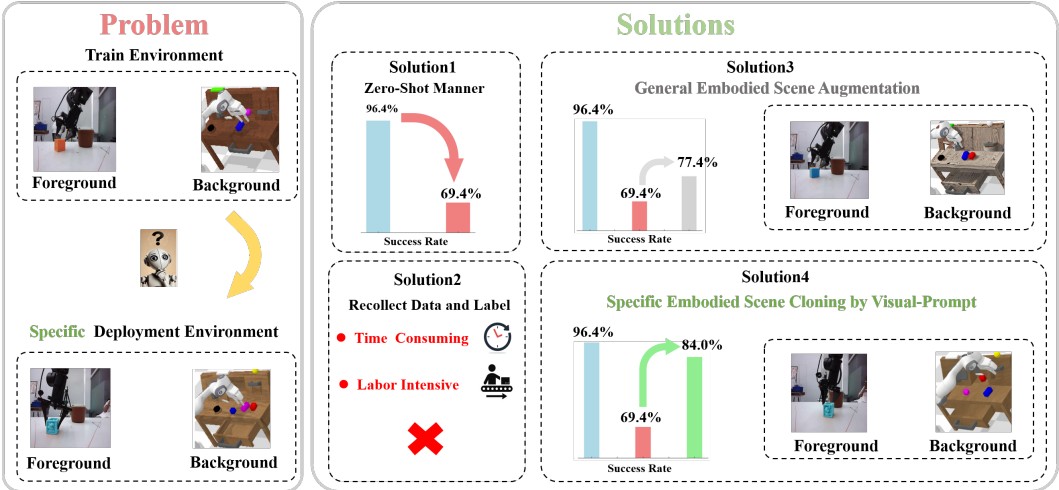

Figure 1: **Left**: Existing embodied policies are often challenged with performance degradation when migrating from a training environment to a deployment environment **Right**: Different solutions are presented. Solution 1 involves direct deployment with zero-shot manner, which results in a significant degradation in performance. Solution 2 entails re-collecting data in the deployment environment to fine-tune the existing method, which is time-consuming and labor-intensive. Solution 3 employs a general embodiment scene augmentation approach conditioned on text prompts. Solution 4 utilizes a specific embodied scene cloning method with visual cues derived from the deployment environment.

Despite recent advancements in developing multi-task generalist policies for robot learning, their generalization capabilities remain far from practical application. Most existing policies are designed for specific embodied scene, where the target tasks and environments closely resemble—or are identical to—those seen during training. As illustrated in solution 1 in Fig.1, adopting one of the current advanced embodied policies, faces a significant performance drop of about 30%—from 96.4% to 69.4% —when deployed in novel embodied scene with unseen foreground and background. Generalizing to unseen objects or environments typically requires additional data collection and model fine-tuning. This falls short of the expected performance, especially considering that the policies, often derived from large-scale vision-language models (VLMs) (Liu et al., 2024; Padalkar et al., 2023), are inherently few-shot or even zero-shot learners. One key factor contributing to these limited generalizations is the insufficient amount of data in the robotics domain. However, scaling up robot learning datasets is particularly challenging due to the labor-intensive and time-consuming nature of real-world data collection. For example, assembling the RT-1 (Brohan et al., 2022) dataset took 17 months and 13 robots to gather just 130,000 demonstrations.

To address generalization challenges, researchers have turned to synthetic methodology. With the rapid advancement of diffusion-based image synthesis techniques, it has become possible to generate high-fidelity visual content from language instructions. In embodied scenarios, methods such as ROSIE(Yu et al., 2023) , GenAug(Chen et al., 2023) and GreenAug(Teoh et al., 2024) have been developed to scale robotic learning with diverse synthetic samples, demonstrating that synthetic data can be highly effective in building generalizable policies.

However, when the goal is to develop a policy for a specific new embodied scene, we could develop a more effective and efficient approach: **Embodied Scene Cloning**. As shown in solution 3 in Fig. 1, existing generative data augmentation methods primarily aim to create diverse samples based on text prompts. However, these methods often struggle to fully capture the detailed characteristics of the deployment environment, such as the detailed textures of specific foreground objects or exact color distribution of specific background. In contrast, embodied scene cloning depicted in solution 4 in Fig. 1, focuses on replicating precise visual samples of the deployment environment, allowing for a more accurate representation of real-world conditions during deployment. Consequently, the synthetic samples are highly effective, closely resembling the deployment environment data, making them as useful as actual data collected from the deployment scenarios. For instance, Embodied Scene Cloning outperformed GreenAug (Teoh et al., 2024), a state-of-the-art text-based generative augmentation method, by increasing the embodied policy's evaluation success rate from 69.4% to 84.0%, representing a 6.6% improvement beyond GreenAug's 77.4%.

The key feature of embodied scene cloning is its ability to enable visual instruction. To ensure that the synthetic data closely mirrors real-world conditions, we developed a novel image editing framework that offers precise control over the visual appearance of generated content. This is accomplished by incorporating external visual objects as conditions during the editing process, along with control mechanisms for specifying the position and shape of these objects. Our visual-instruction guided image editing framework operates in two stages: an initial stage that encodes the input image while preserving key non-edited regions, and a generation stage that focuses on the edited areas. Through a dynamic blending mechanism, the framework ensures smooth transitions between edited and non-edited regions, maintaining the integrity of the original scene while adapting to new visual elements. This allows for the generation of high-fidelity samples that are not only visually accurate but also consistent with the target deployment environment.

We explore how embodied scene cloning can aid in developing generalizable policies in both reproducible simulation environments and practical real-world settings. To demonstrate the effectiveness of our approach, we compare embodied scene cloning with state-of-the-art generative data augmentation methods. Empirical results show that our customized samples are significantly more effective when targeting specific new embodied scene. In summary, our key contributions are:

- We propose a novel data synthesis method called Embodied Scene Cloning, designed to effectively address the generalization challenge in specific novel embodied scene. This approach is distinguished by its visual instruction capability, allowing precise control over both the generated objects and the background environments.

- Experimental results demonstrate significant performance improvements in various language-conditioned visuomotor policies when using our method, showcasing its clear advantages over existing generative augmentation approaches.

## 2 RELATED WORK

### 2.1 LANGUAGE-CONDITIONED EMBODIED POLICIES

Recent advancements in language-conditioned embodied policies have greatly enhanced robots' ability to execute complex tasks by interpreting natural language commands and visual cues(Driess et al., 2023; Shah et al., 2023; Touvron et al., 2023; Radosavovic et al., 2023). Models like CLIPort(Shridhar et al., 2022) and PerAct(Shridhar et al., 2023), leveraging CLIP(Radford et al., 2021), have shown strong performance in robotic manipulation. More recent approaches, such as Hiveformer(Guhur et al., 2023), use transformers to jointly model language, visual history, and multiple views, further improving task completion. Large Language Models (LLMs), like those in SayCan(Ahn et al., 2022) and RT-2(Brohan et al., 2023), translate high-level instructions into executable actions, generalizing across diverse tasks through multimodal fine-tuning. However, generalization remains challenging, with models often struggling in novel environments or with unseen tasks and configurations. Pre-trained models like RoboFlamingo(Li et al., 2023) and RT-2 improve generalization by leveraging large datasets but still face limitations in transferring knowledge to new scenarios.

### 2.2 GENERATIVE DATA AUGMENTATION IN EMBODIED INTELLIGENCE.

Generative models, particularly diffusion models, have advanced robotic learning by enabling synthetic augmentation, reducing reliance on real-world data collection. Early methods like ROSIE(Yu et al., 2023) and GenAug(Chen et al., 2023) used text-to-image models for dataset augmentation. ROSIE employed diffusion-based inpainting to integrate unseen objects and backgrounds, improving generalization. GenAug introduced depth-guided visual diversity, enhancing robustness to new environments. Recent approaches, such as CACTI(Mandi et al.) and GreenAug(Teoh et al., 2024), adopted alternative strategies. CACTI combined generative models with expert data for task-specific augmentations like kitchen object manipulation. GreenAug employs chroma keying to replace green-screen backgrounds with diverse textures and uses generative inpainting in GreenAug-Gen to add realistic scenes like kitchens or living rooms, enhancing visual generalization with a simpler, effective approach. Our visual-prompt-based framework builds on these advancements, focusing on precise adaptation to specific deployment environments, achieving consistent generalization beyond general scene augmentation methods.

### 2.3 VISUAL PROMPTING IN VISION-LANGUAGE MODELS AND GENERATION.

Visual prompting has advanced across domains, with LLAVA(Liu et al., 2024) integrating visual instruction tuning into language models. In robotics, RT-2(Brohan et al., 2023) leveraged visual prompts for task execution, and PaLM-E(Driess et al., 2023) extended these capabilities in embodied agents, VIMA(Jiang et al., 2022) further advanced the field by introducing multimodal prompting, which combines textual and visual tokens to specify tasks for generalist robotic manipulation. While effective, these approaches do not address the challenges of embodied scene transfer beyond semantic understanding. Generative models have explored controlled generation using visual prompts. ControlNet(Zhang et al., 2023) introduced structural conditioning with visual guidance, IP-Adapter(Ye et al., 2023) improved prompt integration, and MS-Diffusion(Wang et al., 2024) enabled layout-preserving multi-content synthesis. However, these methods focus on general image generation, overlooking the unique needs of embodied scenes.

Our method targets three key requirements for embodied scene cloning: (1) **precise control of visual prompt placement** via Grounding-Resampler and cross-attention, (2) **semantic consistency in non-edited regions** using Progressive Masked Fusion to prevent artifacts while preserving task-relevant features, and (3) **trajectory feasibility** through depth-consistent editing with ControlNet, ensuring spatial relationships for valid manipulation sequences. This design enables reliable policy transfer by maintaining visual adaptation, scene consistency, and physical plausibility. For instance, when transferring a grasping task, our method ensures correct object placement, preserves scene context, and maintains valid interaction trajectories—capabilities beyond existing approaches.

### 2.4 IMAGE TRANSLATION FOR ROBOTICS

Recent advancements in image translation for robotics have utilized both GAN-based and diffusion-based methods to address the sim-to-real gap. GAN-based approaches like RL-CycleGAN(Rao et al., 2020) adapted CycleGAN(Chu et al., 2017) with scene consistency losses, ensuring that simulated images retain key semantic and spatial features critical for downstream tasks during domain adaptation. RetinaGAN(Ho et al., 2021) extended this concept by introducing object-detection consistency during GAN training, enforcing invariance in object features between simulated and translated images to ensure reliable domain alignment. Diffusion-based methods, in contrast, focus on precise control over image generation. ALDM-Grasping(Li et al., 2024) employed a layout-to-image diffusion process, enabling photorealistic transformations while preserving spatial layout and object relationships critical for robotic manipulation. Similarly, LucidSim(Yu et al., 2024) utilized depth-conditioned ControlNet to achieve geometrically consistent scene generation, effectively bridging the sim-to-real gap for diverse and dynamic environments.

Unlike these approaches, which primarily target general scene adaptation, our Embodied Scene Cloning method focuses on deployment-specific scene transfer. By leveraging visual prompts from specific deployment environment, it enables precise and efficient adaptation to target environments while maintaining spatial and semantic consistency. This tailored approach allows for more effective scene transfer, addressing challenges in environment-specific deployment.

## 3 METHOD

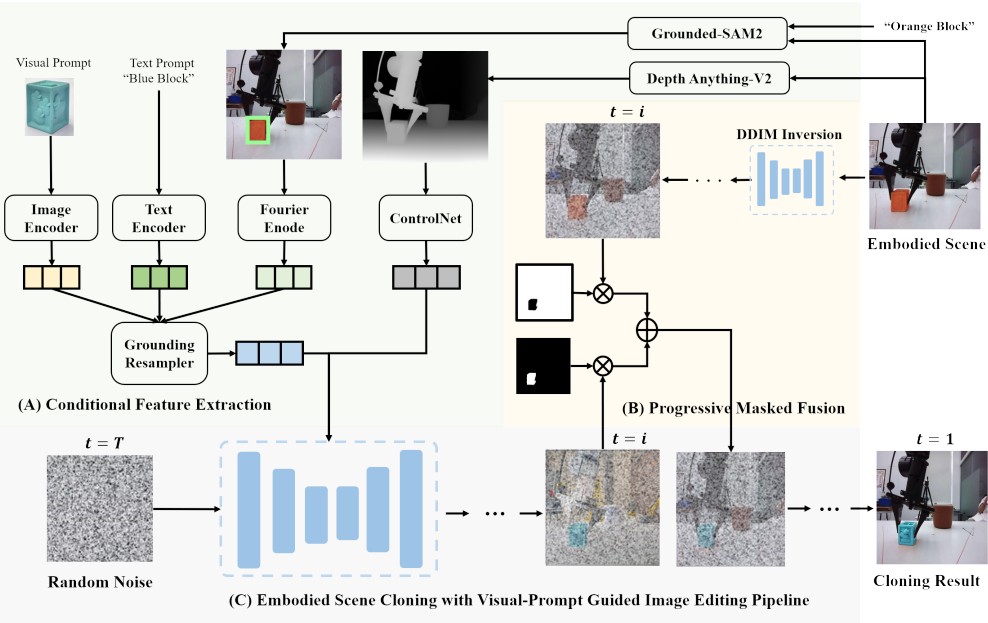

Figure 2: The pipeline for **Embodied Scene Cloning** consists of the following key stages: (a) Conditional Feature Extraction: This stage involves the extraction and processing of conditional features, this stage output including grounded visual prompt embedding and refined depth embedding. (b) Progressive Masked Fusion: This phase consists of how to fuse the diffusion latent stored in the inversion process and the latent from the denosing process by means of an progressive mask strategy in order to enable the generation of new content in the editing region while maintaining the consistency of the content in the non-editing region. (c) Visual-Prompt Guided Image Editing: This phase focuses on injecting the grounded visual prompt embedding and refined depth embedding into the denoising UNet, enabling precise control of the visual prompt to generate at specific locations while maintaining consistency in the non-editing regions.

### 3.1 OVERVIEW OF EMBODIED SCENE CLONING

Unlike traditional augmentation methods that rely on text prompt cues, Embodied Scene Cloning focuses on replicating specific deployment environments by incorporating visual cues from the scene. This approach enables the reuse of existing trajectories, improving the generalization of embodied strategies while achieving more efficient training-free data augmentation. The framework of Embodied Scene Cloning is illustrated in Fig.2. As illustrated in Fig. 2, the process begins by generating the conditional inputs for the subsequent generation stage. Based on the input image and target instructions, we extract the required depth map, segmentation map of the target object, and its coordinate location. These conditional inputs are then processed using ControlNet(Zhang et al., 2023) and GroundingResampler(Wang et al., 2024), deriving the extraction of refined depth embedding and grounded visual prompt embedding. In the next phase, the editing process is divided into two key stages: Inversion and Denoising. Both stages employ a subject-conditioned denoising U-Net(Wang et al., 2024) as the backbone, with the refined depth embedding and grounded visual prompt embedding as condition.

### 3.2 CONDITIONAL FEATURE EXTRACTION

The conditional inputs required for our editing framework include depth maps, edges, and segmentation maps. To extract the depth features, we use DepthAnythingV2(Yang et al., 2024), while Grounded-SAM2(Liu et al., 2023) is employed to generate the bounding boxes and segmentation maps of objects based on locations derived from the text prompts. In the conditional feature processing stage, the visual prompt, text prompt, and the bounding box features of the target object are processed through the CLIP image feature processor(Radford et al., 2021), CLIP text feature processor(Radford et al., 2021), and Fourier encoding, respectively. As a result, we obtain the visual prompt embedding $f_v$, the text prompt embedding $f_t$, and the grounding embedding $f_g$. Additionally, the text prompt embedding and the grounding embedding are used to initialize a query embedding $f_q$, which later interacts with the visual-prompt embedding to inject the coresponding semantic and location information. Next, these embeddings are aggregated using the Grounding-Resampler(Wang et al., 2024) to fuse three types of feature and output the grounded visual-prompt embedding. The attention mechanism within the Resampler is described by the following equation:

$$\text{RSAttn} = \text{Softmax}\left(\frac{\mathbf{Q}(f_q)\mathbf{K}([f_v, f_q])}{\sqrt{d}}\right)\mathbf{V}([f_v, f_q])$$

In this equation, $[f_v, f_q]$ denotes the concatenation of the visual prompt embedding and the query embedding. This attention layer allows the model to enhance the interaction between the visual-prompt and grounding representations.

### 3.3 VISUAL-PROMPT GUIDED IMAGE EDITING WITH PROGRESSIVE MASK FUSION

We propose a visual-prompt-guided image editing model that explicitly divides the editing process into two parts: the reconstruction of non-editing regions and the generation of the editing regions. Specifically, this process is carried out in two stages: the inversion stage and the denoising stage. In the inversion stage, the original image is mapped into the latent diffusion space, and the states at each step are stored for the reconstruction of non-editing regions during the denoising stage. In the denoising stage, we start from a randomly initialized noise and inject grounded visual-prompt embedding along with ControlNet condition features. Using the previously extracted binary mask, the newly generated denoised latent diffusion features at each step are progressively fused with the corresponding latent diffusion features stored in the cache queue. This gradual fusion ensures a harmonious blend between the editing and non-editing regions, ultimately generating an object in the designated image area that matches the visual prompt in a "cloning" manner. In the following, we present a detailed explanation of our design choices in both stages, with prior knowledge of diffusion models and DDIM Inversion provided in Appendix A.1.

**Progressive Masked Fusion.** In Embodied Scene Cloning, the goal is to seamlessly integrate visual elements from a new environment into an existing scene using visual-prompt guided image editing techniques. Whether adapting the foreground or background, it's crucial to preserve non-editing regions to ensure consistency. For instance, if only the background changes while the task remains unchanged, we must maintain the integrity of other areas to prevent interference with task performance,

ensuring smooth adaptation across environments. To achieve this, as shown (B) in Fig.2, in the DDIM Inversion stage, we design a cache queue to store the diffusion latent states at each time step. This cache queue facilitates the accurate reconstruction of non-editing regions and ensures a seamless integration with edited regions during the subsequent denoising phase. By retaining the inverted latent diffusion features across all time steps, the cache queue serves as a reference, allowing non-editing regions to maintain the high-fidelity details of the original environment while still enabling controlled changes in the designated areas. The input raw image feature $I$ is progressively encoded into a series of latent features $\mathbf{z}_t$ at each timestep by the denosing UNet network:

$$\mathbf{z}_t = \text{DDIMInversion}(I, t), \quad t = 1, \ldots, T$$

The stored diffusion latent features $\mathbf{z}_t$ serve as anchor points, allowing for gradual reconstruction during the denoising stage while maintaining coherence between the edited and non-edited parts of the image. In the denoising stage, which starts from random noise, we blend the stored latent features from the inversion stage with the newly generated latents using a progressive mask fusion mechanism. This ensures smooth integration of edited and non-edited regions, preserving the original scene while enabling controlled modifications. The cache queue created during the inversion stage stores the latent features $\mathbf{z}_t$, representing the visual semantic embeddings of the non-edited areas. These stored latent features are fused with the newly generated latent features $\tilde{\mathbf{z}}_t$ via the target mask, and the result of the next denoising step $\tilde{\mathbf{z}}_{t-1}$ is computed using the DDIM forward process:

$$\tilde{\mathbf{z}}_{t-1} = \text{DDIMForward}(M_t \cdot \mathbf{z}_t + (1 - M_t) \cdot \tilde{\mathbf{z}}_t)$$

where $M_t$ controls the blending between the stored and newly generated features. The decay of the mask is modeled as:

$$M_t = M_0 \cdot \left(1 - \frac{t}{T}\right)$$

This progressive decay ensures a smooth transition between edited and non-edited regions, allowing for seamless integration of new scene elements while maintaining the consistency and fidelity of the original, non-edited areas.

**Visual-Prompt Guided Image Editing.** As shown part (C) in Fig.2, In Embodied Scene Cloning, the visual elements of a new environment are injected into the denoising process using a visual-prompt-guided image editing mechanism. The grounded visual-prompt embedding of the novel scene is introduced into the denoising UNet through a cross-attention mechanism, ensuring that the generation of new content is aligned with the provided visual prompt. The visual-grounded features $f_v$, which encapsulate both spatial and semantic information, are integrated with the denoising latents $\tilde{\mathbf{z}}_t$ via cross-attention. At each timestep $t$, the cross-attention mechanism computes the interactions between the visual-grounded features and the denoising latents, enabling the model to focus on and update the specified regions. The cross-attention operation is defined as:

$$\text{Attn}(Q, K, V) = \text{Softmax}\left(\frac{QK^T}{\sqrt{d}}\right) V$$

where: $Q = W_q f_v$ is the query matrix derived from the visual-grounded features, $K = W_k[f_v, \tilde{\mathbf{z}}_t]$ is the key matrix, concatenating the visual-grounded features and the latent features from the current timestep $\tilde{\mathbf{z}}_t$, $V = W_v[f_v, \tilde{\mathbf{z}}_t]$ is the value matrix, containing both the visual-grounded features and the latent features. This cross-attention mechanism enables focused control over the content in the editing regions while maintaining consistency across the scene. The model progressively computes interactions between the visual-grounded features and the denoising latents, allowing for selective updates that conform to the visual cues injected into the model. Following the cross-attention-based injection of visual-prompt embeddings, the edited and non-edited regions are continuously blended in a progressive masked fusion manner across multiple denoising timesteps. This gradual fusion ensures that the integration of new visual elements is both smooth and coherent, maintaining consistency in non-edited regions while allowing controlled modifications in the target areas. By blending features progressively, the model achieves a seamless transition between the original and modified areas, ensuring high-fidelity results.

## 4 EXPERIMENT

### 4.1 IMPLEMENTATION DETAILS

The performance validation of the downstream task is divided into two parts: simulation and real-world testing. For the simulation environment validation, we use the CALVIN benchmark, which focuses on long-horizon, language-conditioned robotic tasks. The benchmark includes four distinct manipulation environments, each featuring a robotic arm interacting with objects like drawers and sliders. It supports 34 unique tasks and challenges agents to follow sequential language instructions to complete complex manipulation actions. As policy learning methods, we adopt two advanced methods from the CALVIN benchmark: RoboFlamingGo(Li et al., 2023) and 3D Diffusion Actor(Ke et al., 2024). For the real-world experiments, we employ the Aloha setup and use LLAVA(Liu et al., 2024) as the policy learning methods for end-to-end action prediction. For each real-world experimental condition, we conducted 10 trials and reported the average success rate, which is consistent with previous research(Kim et al., 2024; Black et al., 2024; Team et al., 2024).

### 4.2 MAIN RESULTS.

We design experiments to answer the following research questions: (1) Does the novel foreground generated by Embodied Scene Cloning provide sufficient enhancement? (2) Does Embodied Scene Cloning result in a significant performance improvement for overall task completion? (3) How does the performance of Embodied Scene Cloning scale with increasing amounts of cloned data for tasks in the new environment? (4) Does Embodied Scene Cloning show consistent improvement across different state-of-the-art policy learning methods? (5) Does Embodied Scene Cloning still provide performance gains in real-world environments?

**Experiment Setup.** To address this question, we introduced a new experimental setup on the Calvin benchmark, labeled [ABC] -> D. Here, as shown in Fig.3, ABC refers to three distinct environments used for training, while D represents a novel test environment with an unseen background. Unlike the standard ABC -> D setup, where the model is tested in a new scene D after training on ABC, our [ABC] -> D setup introduces an additional challenge. Specifically, we excluded all trajectories related to the pink block tasks from the training data in ABC and then evaluated the model on environment D, which includes tasks involving red, blue, and pink blocks (e.g., "Lift red/blue/pink block from the table"). This setup helps assess the model's generalization to both unseen environments and unseen objects. Based on this setting, we tested the impact of Embodied Scene Cloning and GreenAug, comparing their ability to clone the novel foreground or background to existing training trajectories in [ABC]. This allowed us to evaluate whether visual-prompt based scene cloning outperforms traditional text-prompt based augmentation methods.

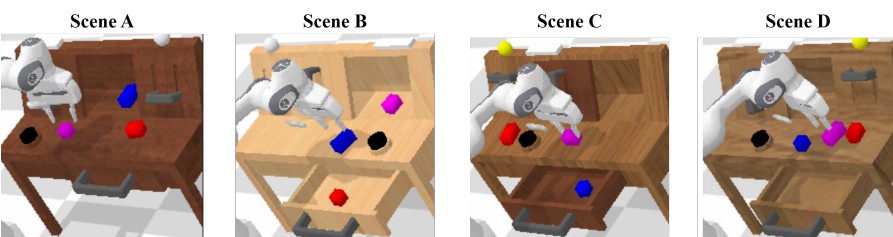

Figure 3: The visualization of the four embodied scene demonstration in CALVIN.

(1) **Embodied Scene Cloning with Novel Foreground.** To answer the first question, we used Embodied Scene Cloning to modify the training trajectories involving blue and red blocks. By injecting pink blocks as visual prompt into these existing trajectories, we created new scene-cloning data, termed Embodied Scene Cloning-FG. In comparison, we applied GreenAug's generative inpainting method, using the text prompt "pink block" to enhance the original training trajectories related to blue or red block, resulting in GAug-FG. As shown in the results from Table 1, the synthetic data generated by Embodied Scene Cloning effectively improved the model's generalization ability to unseen foreground tasks. The generated data from Embodied Scene Cloning outperformed GreenAug by approximately 18.6% (65.10% vs. 46.49%) in tasks involving new objects. This indicates that

cloning-based injection with visual prompt is more effective at enhancing policy generalization to new scenes than existing general data augmentation methods rely on text-prompt.

Table 1: RoboFlamingGo imitation performance across different task splits (Train → Test) with varying data augmentations. The columns lift_slider, rotate_left, rotate_right, lift_table, push_left, push_right, and lift_drawer represent the success rates for tasks involving manipulation of the pink block. GAug refers to a generative augmentation algorithm used in GreenAug, while Embodied Scene Cloning denotes our proposed method for generative augmentation. Bold values highlight the best performance for each task.

| RoboFlamingGo (Train → Test) | lift_slider | rotate_left | rotate_right | lift_table | push_left | push_right | lift_drawer | Avg |
|---|---|---|---|---|---|---|---|---|
| [ABC] → D | 1.0 | 1.9 | 1.9 | 2.5 | 13.0 | 14.0 | 33.3 | 9.68 |
| [ABC] + GAug-FG → D | 30.1 | 55.9 | 21.2 | 40.9 | 54.5 | 47.8 | 75.0 | 46.49 |
| [ABC] + Embodied Scene Cloning-FG → D | **47.9** | **77.5** | **55.2** | **58.5** | **71.0** | **62.3** | **83.3** | **65.10** |

(2) **Overall Task Improvements.** To address the second research question, we first evaluated the impact of Embodied Scene Cloning on synthetic foregrounds across all tasks in the Calvin benchmark. By visually injecting new foregrounds, Embodied Scene Cloning significantly improved overall task performance. As shown in Table 2, task 1 saw a 14.6% improvement, while task 5 had an 8.0% gain, both outperforming GreenAug, which relies on text-prompt conditioning. GreenAug achieved a 11.9% improvement in task 1 and a 7.8% increase in task 5, highlighting Embodied Scene Cloning's superior performance with synthetic foreground samples. For background data augmentation, Embodied Scene Cloning utilized the table from the test environment D as a visual prompt to modify the backgrounds of the original training trajectories, resulting in Embodied Scene Cloning-BG data. In comparison, GreenAug applied text-based augmentation using prompts such as "wooden table" to modify the backgrounds, generating GAug-BG data. Across all tasks, Embodied Scene Cloning-BG demonstrated notable performance improvements, with a 14.6% increase in task 1 and an 8.0% increase in task 5. Additionally, when focusing specifically on novel backgrounds, Embodied Scene Cloning-BG continued to outperform GreenAug, achieving a 6.5% increase in task 1 and a 2.4% boost in task 5, while GreenAug's gains were 6.8% and 1.1%, respectively. Lastly, when combining both synthetic foregrounds and backgrounds, Embodied Scene Cloning demonstrated even more significant advantages. The combined effects of foreground-cloning and background-cloning produced higher performance gains than either category alone, resulting in a compounded benefit that further outperformed GreenAug's mixed samples.

Table 2: RoboFlamingGo imitation performance across different task splits (Train → Test) with varying data augmentations. The table shows success rates across five tasks (1 to 5), as well as the average sequence length (AvgLen). GAug refers to generative augmentation methods applied to the foreground (FG) and background (BG). Bold values highlight the best performance for each task.

| RoboFlamingGo (Train → Test) | 1 | 2 | 3 | 4 | 5 | AvgLen |
|---|---|---|---|---|---|---|
| [ABC] → D | 69.4 | 45.9 | 30.1 | 20.3 | 13.2 | 1.79 |
| [ABC] + GAug-FG → D | 72.1 | 46.8 | 29.7 | 20.2 | 13.4 | 1.82 |
| [ABC] + Embodied Scene Cloning-FG → D | **84.0** | **63.4** | **44.8** | **31.2** | **21.2** | **2.45** |
| [ABC] + GAug-BG → D | 69.1 | 46.2 | 31.0 | 22.3 | 14.5 | 1.83 |
| [ABC] + Embodied Scene Cloning-BG → D | **75.9** | **51.6** | **35.3** | **24.7** | **15.6** | **2.03** |
| [ABC] + GAug-FG-BG → D | 77.4 | 52.8 | 36.7 | 23.4 | 14.6 | 2.05 |
| [ABC] + Embodied Scene Cloning-FG-BG → D | **84.0** | **65.1** | **47.4** | **35.2** | **25.6** | **2.57** |
| ABC → D | 82.4 | 61.9 | 46.6 | 33.1 | 23.5 | 2.48 |
| ABC + Embodied Scene Cloning-BG → D | **86.6** | **69.6** | **53.2** | **41.8** | **32.6** | **2.83** |

(3) **Scaling trends in cloned synthetic samples by Embodied Scene Cloning.** To answer the third question, we did the experiment in Table 3 below. As can be seen in Table 3, as the amount of data goes from 0.1k to 0.5k to 0.9k, the samples generated by Embodied Scene Cloning bring consistent performance gains in new scenarios, fully demonstrating the scalability of the samples and the potential of the method.

Table 3: RoboFlamingGo imitation performance across different task splits (Train → Test) with varying Embodied Scene Cloning dataset sizes. The table shows the success rates for tasks involving manipulation of the pink block. The last column (Avg) represents the average success rate across all tasks. Bold values highlight the best performance for each task.

| RoboFlamingGo (Train → Test) | lift_slider | rotate_left | rotate_right | lift_table | push_left | push_right | lift_drawer | Avg |
|---|---|---|---|---|---|---|---|---|
| [ABC] + Embodied Scene Cloning (0.1K) → D | 43.9 | **78.4** | 23.4 | **73.1** | 48.1 | 47.9 | 66.7 | 54.5 |
| [ABC] + Embodied Scene Cloning (0.5K) → D | 47.9 | 77.5 | 55.2 | 58.5 | 71.0 | 62.3 | **83.3** | 65.1 |
| [ABC] + Embodied Scene Cloning (0.9K) → D | **82.9** | 60.0 | **75.0** | 64.7 | **89.7** | **63.3** | 55.2 | **70.1** |

(4) **Performance improvement of Embodied Scene Cloning on different policy learning methods.** In order to verify whether Embodied Scene Cloning has sufficient performance gain for different policy learning methods, as shown in Table.4, we tested another state-of-the-art approach, 3d-diffusion-actor, and the experimental results show that 3d-diffusion-actor still faces challenges when migrating in embodied environments and has a significant performance gain after using Embodied Scene Cloning data, with 9.4% improvement in stage1 and 12.5% improvement in stage5, which demonstrates that embodiment cloing can work well on different advanced policy learning methods.

Table 4: Comparison of RoboFlamingGo and 3D Diffusion Actor imitation performance across different task splits (Train-Test). The table shows success rates across five tasks (1 to 5), as well as the average sequence length (Avg). Embodied Scene Cloning-FG refers to applying the Embodied Scene Cloning method with foreground augmentation. Bold values highlight the best performance for each method across tasks.

| Method | Train-Test | 1 | 2 | 3 | 4 | 5 |
|---|---|---|---|---|---|---|
| RoboFlamingGo | ABC → D | 82.4 | 61.9 | 46.6 | 33.1 | 23.5 |
| RoboFlamingGo | [ABC] → D | 69.4 | 45.9 | 30.1 | 20.3 | 13.2 |
| RoboFlamingGo | [ABC] +Embodied Scene Cloning-FG → D | **84.0** | **63.4** | **44.8** | **31.2** | **21.2** |
| 3D Diffusion Actor | ABC → D | 92.2 | 78.7 | 63.9 | 51.2 | 41.2 |
| 3D Diffusion Actor | [ABC] → D | 75.4 | 52.8 | 35.6 | 24.2 | 17.1 |
| 3D Diffusion Actor | [ABC] + Embodied Scene Cloning-FG → D | **84.8** | **66.7** | **52.5** | **40.6** | **29.6** |

(5) **Performance gains from Embodied Scene Cloning in real-world scenarios.** To evaluate the effectiveness of Embodied Scene Cloning in real environments, as shown in Table 5, we designed a task where a robotic arm clamps cubes from a table and places them inside a basket. Initially, we trained an LLM on robotic arm grasping data related to yellow cubes, achieving a 60% success rate in task performance. However, when the model was transferred to a similar task involving blue cubes, the success rate dropped to 0%. After augmenting the training data with a mix of original and Embodied Scene Cloning data, the success rate for grasping blue cubes improved to 20%. These results suggest that Embodied Scene Cloning holds significant potential for real-world applications.

Table 5: Comparison of LLM performance across different train-test settings. The table shows success rates when transferring tasks between different block colors. Embodied Scene Cloning-FG is applied for training on YellowBlock and testing on BlueBlock.

| LLM (Train → Test) | Success Rate |
|---|---|
| YellowBlock → YellowBlock | 60% |
| YellowBlock → BlueBlock | 0% |
| YellowBlock + Embodied Scene Cloning-FG → BlueBlock | 20% |

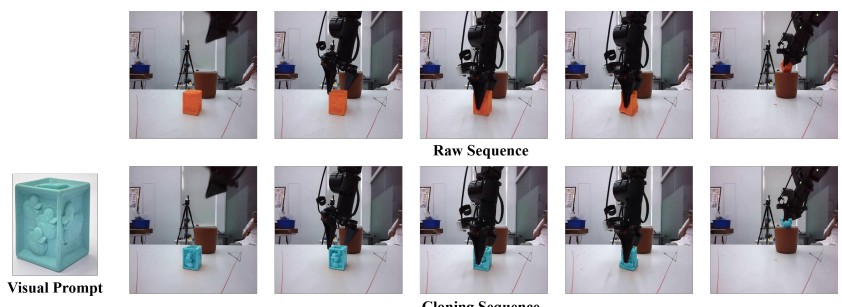

Figure 4: Visualisation results of Embodied Scene Cloning with novel foreground.

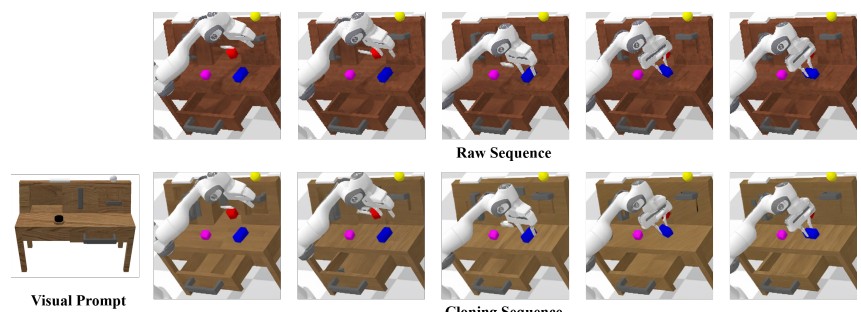

Figure 5: Visualisation results of Embodied Scene Cloning with novel background.

### 4.3 VISUALISATION RESULTS

**Embodied Scene Cloning with novel foreground.** As shown in Fig.4, Embodied Scene Cloning ensures consistency between the edited and non-edited regions while accurately injecting visual concepts into the foreground. This process allows for seamless cloning of novel foreground elements from the new embodiment (i.e., the novel objects to be manipulated) into the existing scene, while preserving any pre-existing deformations. For instance, the fourth and fifth columns in Fig. 3 demonstrate this effect.

**Embodied Scene Cloning with novel background.** As illustrated in Fig.5, Embodied Scene Cloning effectively incorporates a new visual background (wodden table) into the existing scene. The cloned background aligns more accurately with the specific visual features of the cue, resulting in a coherent and visually realistic scene.

## 5 CONCLUSION

In this paper, we presented Embodied Scene Cloning, a novel visual-prompt-based framework designed to address the challenge of generalizing robotic policies to specific deployment embodied scene. By leveraging visual cues from the target environment, Embodied Scene Cloning generates visual-aligned trajectories, enabling effective "cloning" of source demonstrations into the target environment. This approach minimizes the need for costly and time-consuming data collection and annotation in new environments. Unlike traditional methods that foucs on general scene data augmentation, our framework focuses on visual-grounded signals, allowing for more accurate and coherent adaptation. Our experimental results demonstrate that Embodied Scene Cloning significantly improves policy generalization in both simulated and real-world settings, providing substantial performance gains across a variety of tasks. This highlights the potential of our method as an effective tool for addressing environmental discrepancies in embodied intelligence tasks. Moving forward, Embodied Scene Cloning offers a promising approach to embodied data augmentation, paving the way for more efficient and scalable solutions in robotic learning and deployment.

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

# A APPENDIX

## A.1 PRELIMINARIES OF DIFFUSION MODELS AND DDIM INVERSION

Diffusion models generate data by progressively transforming random noise into a structured output, such as an image, over a series of timesteps. The process begins with a noise vector $z_T$, and by gradually denoising through multiple steps, the model recovers the original latent representation $z_0$. During the training phase, the model learns to predict and remove the noise added at each timestep. At time $t$, the noisy data $z_t$ is generated using the following equation:

$$z_t = \sqrt{\alpha_t}z_0 + \sqrt{1 - \alpha_t}\epsilon$$

where $\epsilon \sim \mathcal{N}(0, 1)$ represents Gaussian noise, and $\alpha_t$ is a predefined hyperparameter controlling the noise schedule. The network $\epsilon_\theta$ is trained to predict the noise $\epsilon$ using an $L_2$-based loss objective. The denoising process occurs iteratively, and at each step, the model predicts $z_{t-1}$ from $z_t$ using:

$$z_{t-1} = \frac{\sqrt{\alpha_{t-1}}}{\sqrt{\alpha_t}}z_t + \left(\sqrt{1 - \alpha_{t-1}} - \sqrt{1 - \alpha_t}\right)\epsilon_\theta(z_t, t)$$

By iterating this equation over several timesteps, the model gradually removes the noise and retrieves the clean data $z_0$.

DDIM(Song et al.) provides a deterministic alternative to traditional diffusion models by enabling the inversion of the diffusion process. This allows the model to map an image or data representation $z_0$ back to a noisier state $z_T$ while preserving the structure of the latent space. The inversion from $z_t$ to $z_{t+1}$ can be performed using a similar formulation as the denoising process:

$$z_{t+1} = \frac{\sqrt{\alpha_{t+1}}}{\sqrt{\alpha_t}}z_t + \left(\sqrt{1 - \alpha_{t+1}} - \sqrt{1 - \alpha_t}\right)\epsilon_\theta(z_t, t)$$

This enables applications such as reconstructing latent representations for real images and editing, where precise control over the generative process is necessary. The reversibility of DDIM is grounded in treating the denoising process as a differential equation, allowing the model to operate efficiently both forwards and backwards.

## A.2 MORE VISUALIZATION

In this section, we present further visualizations to demonstrate the application of our method in various embodied environments, including our real-world validation scene, RT1/RT2 scene, and the Behavior-1K environments. The results highlight the robustness of our approach in multiple and cross-category visual prompt cloning. Additionally, they showcase the method's potential for simulation-to-reality transfer, further validating its versatility and effectiveness.

**Visualization in more datasets (RT1/RT2/Behavior-1K).** We apply Embodied Scene Cloning to robotic demonstrations from RT1,RT2 and Behavior-1K to showcase its capability in broader category visual prompt cloning, extending beyond simple cloning of color and texture. As illustrated in Fig.6, our method effectively transfers object trajectories to items with distinct appearances while maintaining seamless integration with the background. Examples include transferring a trajectory from a soda can to a double-walled glass cup, a football to a volleyball. The visualization of the experimental results demonstrates the ability of our method to generalize to a broader range of categories in diverse scenes.

**Visualization for multi visual-prompt cloning capability.** Fig.7 and .8 demonstrate how Embodied Scene Cloning integrates multiple distinct visual prompts into various objects and backgrounds within a scene by binding them to their positional signals. The visualization results show that our method achieves realistic cloning.

**Visualization in our real-world validation scene.** Fig.9 and 10 provide additional visualization results of our method in real-world validation scene, showcasing visual scene transfer within the same category and across different categories. They also illustrate the performance of original and cloned foregrounds under varying initial positions and grasping postures. The results demonstrate that our method is not limited to cloning textures and colors but can also handle moderate size variations effectively.

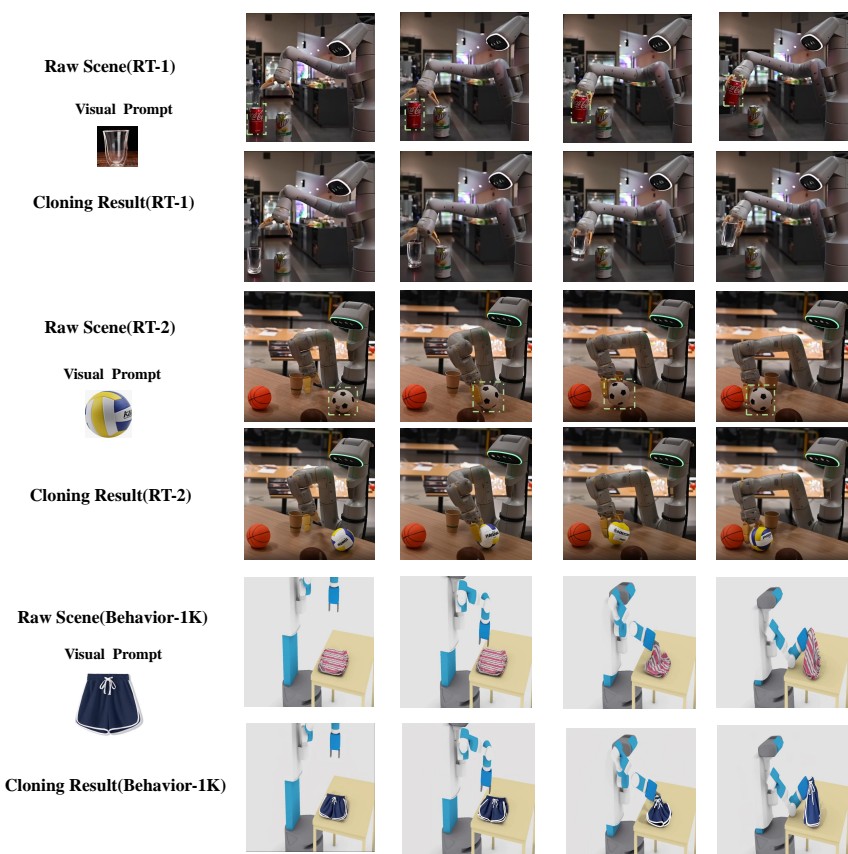

Figure 6: Visualisation results of applying Embodied Scene Cloning to demonstrations from RT1,RT2 and Behavior-1K to show its broader visual prompt cloning capability.

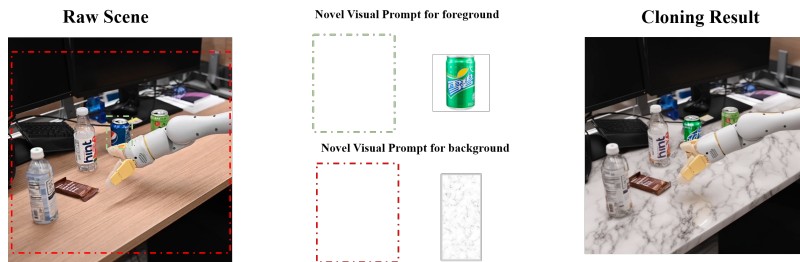

Figure 7: The visualization results of injecting novel foreground/background visual prompts into the scene through Embodied Scene Cloning.

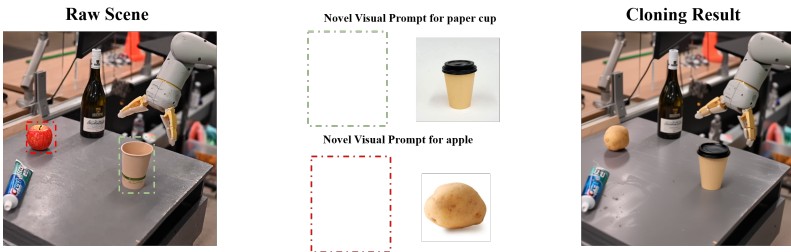

Figure 8: The visualization results of injecting multi novel visual prompts into the scene through Embodied Scene Cloning.

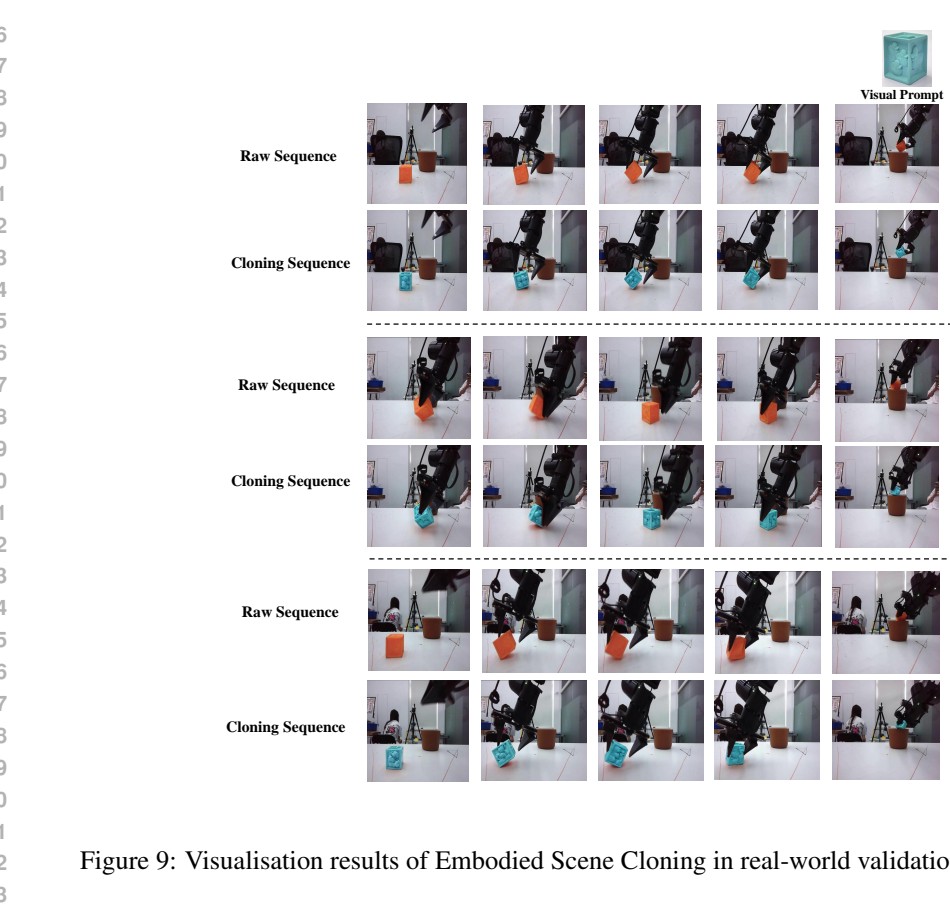

Figure 9: Visualisation results of Embodied Scene Cloning in real-world validation experiments.

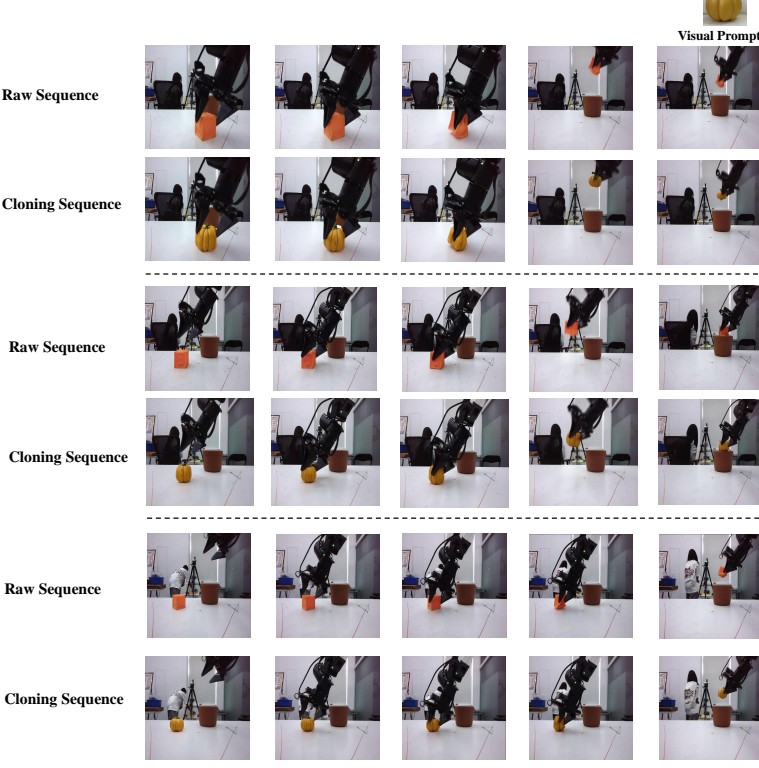

Figure 10: Visualisation results of Embodied Scene Cloning in real-world validation scene.

**Visualization for sim-to-real transfer potential.** To demonstrate the potential of Embodied Scene Cloning for simulation-to-reality transfer, we applied the method to robotic demonstrations in the Behavior-1K. As shown in the first column on the left in Fig.11, the cloning results illustrate that real-world visual appearance prompts significantly enhanced the realism of the simulated scenes, reducing the gap between simulation and reality and showcasing the method's potential for simulation-to-reality scene transfer.

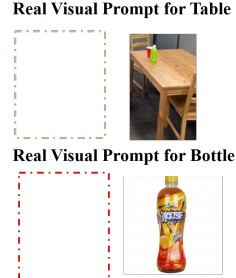
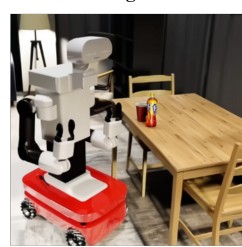

Figure 11: Visualisation results of applying Embodied Scene Cloning to demonstrations from Behavior-1K to show its sim-to-real transfer potential.

### A.3  ABLATION STUDY

We conduct ablation studies to analyze two key components in our framework: Progressive Masked Fusion (PMF) and Visual-Prompt Guided (VPG) Image Editing. As shown in Table 6, PMF plays a crucial role in maintaining coherence between edited and non-edited regions through a mask-guided blending strategy. Without PMF, the non-edited regions may suffer from out-of-distribution (OOD) artifacts, leading to a performance drop from 2.45 to 1.89 in Avglen. For VPG, we introduce an attention scaling factor $\gamma$ that modulates the strength of cross-attention between visual prompts and generated content, similar to (Wang et al., 2024) where $\gamma$ controls how strongly the conditions influence the generation process. A larger $\gamma$ (0.6) ensures stronger attention to visual prompts and better semantic alignment in edited regions, while smaller values (0.2, 0.3) lead to insufficient visual control and degraded performance. The optimal configuration combines PMF with $\gamma$=0.6, demonstrating the importance of both maintaining non-edited region consistency and strong visual prompt guidance.

Table 6: Ablation studies on Progressive Masked Fusion (PMF) and attention scaling factor $\gamma$ in Visual-Prompt Guided Image Editing (VPG).

| PMF | | VPG | | | 1 | 2 | 3 | 4 | 5 | Avglen |
|---|---|---|---|---|---|---|---|---|---|---|
| w | w/o | $\gamma$=0.2 | $\gamma$=0.3 | $\gamma$=0.6 | | | | | | |
| ✓ | | ✓ | | | 71.7 | 45.1 | 29.7 | 20.5 | 12.3 | 1.79 |
| ✓ | | | ✓ | | 73.5 | 49.1 | 33.1 | 23.6 | 15.2 | 1.95 |
| ✓ | | | | ✓ | **84.0** | **63.4** | **44.8** | **31.2** | **21.2** | **2.45** |
| | ✓ | | | ✓ | 72.6 | 48.9 | 31.3 | 22.0 | 14.8 | 1.89 |

### A.4  FAIL CASE ANALYSIS.

Fig.12 illustrates a failure case of our approach. The objective was to clone the blue cube from the target scene into the position of the orange block in the original scene. However, GroundedSAM2 incorrectly identified the dark orange basket in the background as the target orange block, leading to an erroneous mask and subsequently incorrect visual prompt injection. We anticipate that ongoing advancements in open-domain detection models and multimodal models will mitigate these types of limitations in the future.

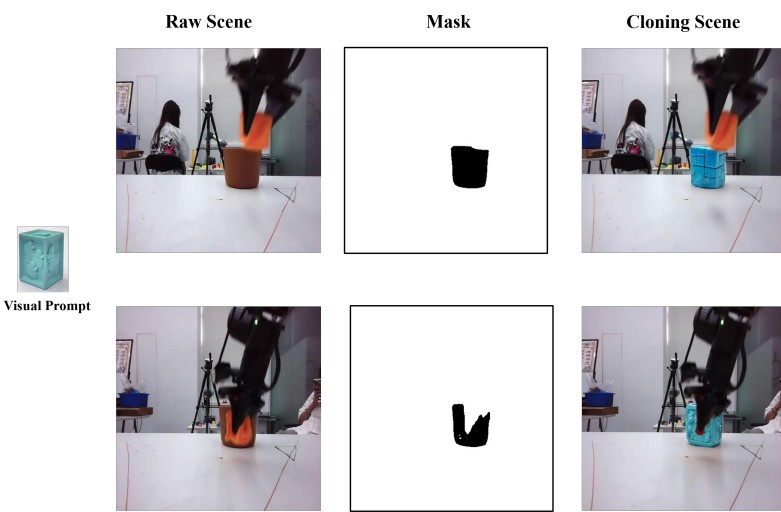

Figure 12: Failure case of Embodied Scene Cloning.

## A.5 COMPARISON WITH TRADITIONAL AUGMENTATION.

To evaluate the effectiveness of traditional image augmentation methods, we implemented color jitter with standard parameters (brightness=±0.4, contrast=±0.4, saturation=±0.4, hue=±0.1). While color jitter shows some improvement over the zero-shot setting, its low-level pixel perturbations fail to address the core challenge of scene generalization in embodied tasks. Unlike these pixel-level transformations, Embodied Scene Cloning explicitly focuses on the semantic adaptation between training and deployment environments through visual-prompt guided control. This is evidenced by the significant improvement in average sequence length (2.57 vs 1.88), indicating our method's superior capability in handling complex, multi-step tasks.

Table 7: Comparison of different augmentation methods on CALVIN benchmark. We compare the performance of zero-shot transfer, traditional color jitter augmentation and our proposed Embodied Scene Cloning method.

| Augmentation Setting | 1 | 2 | 3 | 4 | 5 | Avglen |
|---|---|---|---|---|---|---|
| [ABC] → D (Zero Shot) | 69.4 | 45.9 | 30.1 | 20.3 | 13.2 | 1.79 |
| [ABC] → D (w Color Jitter) | 73.8 | 49.5 | 32.1 | 19.7 | 12.6 | 1.88 |
| [ABC] → D (w Embodied Scene Cloning-FG-BG) | **84.0** | **65.1** | **47.4** | **35.2** | **25.6** | **2.57** |

