# OpenReview forum: "Embodied Scene Cloning: Solving Generalization in Embodied AI via Visual-Prompt Image Editing"
_ICLR.cc/2025/Conference — ICLR 2025 Conference Withdrawn Submission_

### Official Review · Reviewer_XU12 · 2024-10-30

**Soundness:** 3
**Presentation:** 2
**Contribution:** 2
**Rating:** 5
**Confidence:** 4

**Summary:**

The paper introduces a visual-prompt-based framework that enhances policy generalization in robotics by prompting with visual cues from deployment environments. It clones scenes with visual alignment, improving performance in novel environments.

**Strengths:**

Originality: applying diffusion based image editing to change foreground background in robotics
Quality: good ablation study on elements of environment
Clarity: explaining context methods
Significance: improve policy domain transfer

**Weaknesses:**

The main weakness is the tested environment is relatively simple. Only changing the color/texture of one object in foreground / background. To demonstrate real potential, needs more extensive experiments.

**Questions:**

1. The papers refers "traditional augmentation methods" as text based. Meanwhile, there are other traditional augmentation such as color jitter, random crop etc. Suggest directly refer them as text based. In fact, it would be nice see if simple color jitter improve the performance.
2. In line 323, it says including baselines of RoboFlamingGo(Li et al., 2023) and 3D Diffusion Actor. Meanwhile, they are policy learning method not augmentation method. Therefore, it's confusing to call them baselines (which should be GreenAug for example).
3. More papers in image translation for robotics should be discussed in related works. For example:
Li, Yiwei et al. “ALDM-Grasping: Diffusion-aided Zero-Shot Sim-to-Real Transfer for Robot Grasping.” ArXiv abs/2403.11459 (2024): n. pag.

---

> ### Author Response · Authors · 2024-11-27
>
> We sincerely thank Reviewer XU12 for their careful review of our work. We deeply appreciate your recognition of the paper's originality in applying diffusion-based image editing to robotics, the clarity in explaining contextual methods, and the good ablation study. Your acknowledgment of our contribution to improving policy domain transfer is particularly encouraging. Below, we address your comments point by point.
>
> **W1: Relatively simple tested setting**
>
> **A1:** Thank you for the suggestion. To further demonstrate the effectiveness of our method, we have extended the experimental scenarios in the following ways:
> 1. **More datasets**: Figure 6, now includes results on RT-1, RT-2, and Behavior-1K, showcasing our method's performance in more challenging and diverse scenes.
> 2. **More complex settings**: Figure 7 features experiments where both the object and the background are simultaneously modified, highlighting the advanced capabilities of our approach.
> 3. **Beyond color/texture modifications**: Figure 10 includes an example where the box is replaced with a pumpkin, illustrating the flexibility of our method in modifying object shapes.
>
> We would like to emphasize that embodied scene cloning is a general editing framework applicable to a wide range of scenarios. However, due to the fact that robotic manipulation experiments are inherently challenging to conduct fairly due to reliance on proprietary hardware, we chose to present our main results using a reliable simulator alongside a real-world case study. We believe these results sufficiently demonstrate the effectiveness of embodied scene cloning.
>
> **Q1: Comparison with color jitter augmentation**
>
> **A2:** We have included results for color jitter augmentation in Table 7, which we also provide below for convenience. The results demonstrate that our method significantly outperforms color jitter, with an average sequence length of 2.57 compared to 1.88 for color jitter.
> | Augmentation Setting                             | 1    | 2    | 3    | 4    | 5    | Avglen |
> |--------------------------------------------------|------|------|------|------|------|--------|
> | [ABC] → D (Zero Shot)                            | 69.4 | 45.9 | 30.1 | 20.3 | 13.2 | 1.79   |
> | [ABC] → D (w Color Jitter)                       | 73.8 | 49.5 | 32.1 | 19.7 | 12.6 | 1.88   |
> | [ABC] → D (w Embodied Scene Cloning-FG-BG)       | **84.0** | **65.1** | **47.4** | **35.2** | **25.6** | **2.57**   |
>
> **Q2: Confusing terminology "baseline" in Line 323**
>
> **A3:** Thank you for the suggestion. We have revised the terminology, replacing "baseline" with "policy learning methods" for improved clarity.
>
> **Q3: Add related works**
>
> **A4:** We have expanded the related works section by including a discussion on image translation methods for robotics in Section 2.4 of the revised manuscript.

---

> > ### Author Response · Authors · 2024-12-02
> > **Looking forward to discussing**
> >
> > Dear Reviewer XU12
> >
> > Thank you again for your insightful review. We have submitted point-by-point responses to your questions and concerns. We trust that our responses have satisfactorily resolved your concerns. With the discussion phase deadline approaching, we would greatly appreciate it if you could let us know if you have any additional questions. We are happy to respond as soon as possible.

---

### Official Review · Reviewer_i9yN · 2024-11-01

**Soundness:** 3
**Presentation:** 3
**Contribution:** 2
**Rating:** 6
**Confidence:** 3

**Summary:**

This paper proposes a novel data augmentation method called Embodied Scene Cloning for addressing the generalization challenge in robotic manipulation. The proposed method incorporates Conditional Feature Extraction, Progressive Masked Fusion, and Visual-Prompt Guided Image Editing to fulfill visual-instructed source demonstration cloning. As a result, the proposed method allows precise visual control over both the generated objects and the background environments compared to text-prompt-based data augmentation methods. Experimental results in both simulation and real-world environments demonstrate the superiority of the proposed approach over the state-of-the-art generative data augmentation methods, especially GreenAug.

**Strengths:**

1 The proposed approach provides a potential solution to seamlessly integrate of new scene elements while maintaining the consistency and fidelity of the background areas. This will inspire the community to design visually-controlled data augmentation strategies in the future.

2 The effectiveness of the proposed approach is demonstrated in both simulation and real-world environments.

**Weaknesses:**

1 My main concern is the actual generalization ability of the proposed method in real-world applications and the fairness of comparing the proposed method with existing methods. Generally, it is defaulted that the agent in the trained environment cannot access any information in unseen environments. And augmentation for a specific new embodied scene may not be a scalable way of generalization since the unseen scenarios are infinite. Moreover, the proposed policy seems to be relatively effective for texture or color generalization for objects, but its generalization to more common situations such as objects of different types, stacks, and occlusions is not verified.

2 Additionally, the proposed method mainly focuses on the robotic manipulation task while the title claims a large topic by “Solving Generalization in Embodied AI”. I suggest the author give a more precise claim in both the title and the main text.

3 There lack the ablation studies to analyze the effect of specific design for different method components. Moreover, the failure cases should be given for a more complete analysis of the proposed approach.

4 Can the proposed method support sim2real augmentation? The additional experiments on this are appreciated. It will further demonstrate the generalization ability and practicality of the proposed approach.

5 Visual prompting is a widely used strategy in both VLMs and text-to-image generation approaches. The authors should add an independent subsection to analyze existing methods and discuss the comparison in the Related Work section.

**Questions:**

See details in the weaknesses.

---

> ### Author Response · Authors · 2024-11-27
>
> We sincerely thank Reviewer i9yN for the insightful feedback. We deeply appreciate your recognition that our approach 'provides a potential solution to seamlessly integrate new scene elements' and that it will 'inspire the community.' Below, we provide point-by-point responses.
>
> **W1: Generalization ability**
>
> **A1**: Our method is designed for deployment scenarios where data collection has not yet occurred, a common real-world case. In such scenarios, access to the deployment environment is assumed, and our method can significantly enhance performance at a minimal cost compared to conducting full data collection.
> It is important to note that our approach does not aim to improve generalization across infinitely unseen scenarios. While this remains the ultimate goal of data augmentation, existing solutions have yet to achieve this and often struggle to perform effectively in specific deployment environments, as shown in Table 1.
>
> **W2: Focusing on manipulation tasks**
>
> **A2**:Thank you for the suggestion. We have updated the terminology to "Robotic Manipulation" in both the title and the main text for greater clarity.
>
> **W3: Lack of ablation studies**
>
> **A3**:We have added ablation experiments focusing on the use of progressive masked fusion and key hyperparameters in visual prompt-guided image editing. The results, shown in the table below, demonstrate the effectiveness of our design. These experiments are also included in Appendix A.3 of the revised manuscript.
> | PMF (w) | PMF (w/o) | VPG (γ=0.2) | VPG (γ=0.3) | VPG (γ=0.6) |   1    |   2    |   3    |   4    |   5    | Avglen |
> |---------|-----------|-------------|-------------|-------------|--------|--------|--------|--------|--------|--------|
> |    ✓    |           |      ✓      |             |             |  71.7  |  45.1  |  29.7  |  20.5  |  12.3  |  1.79  |
> |    ✓    |           |             |      ✓      |             |  73.5  |  49.1  |  33.1  |  23.6  |  15.2  |  1.95  |
> |    ✓    |           |             |             |      ✓      | **84.0** | **63.4** | **44.8** | **31.2** | **21.2** | **2.45** |
> |         |     ✓     |             |             |      ✓      |  72.6  |  48.9  |  31.3  |  22.0  |  14.8  |  1.89  |
>
>
>
>
> **W3.1: Lack of failure case analysis**
>
> **A3.1**:We have included visualizations of failure cases in Fig.12 ( Appendix A.4 ) of the revised manuscript. A typical failure arises from inaccuracies in the mask generated by GroundedSAM2.
>
> **W4: Support for sim-to-real augmentation**
>
> **A4**:Thank you for this insightful suggestion. We believe our method is applicable to sim-to-real scenarios by combining real-world visual appearances with simulated trajectories. Visualization results have been added to Fig. 11 of the revised manuscript. However, due to resource limitations, we were unable to conduct real-world sim-to-real experiments and have identified this as an important direction for future work.
>
> **W5: Discussion of visual prompting**
>
> **A5**:We have expanded the discussion on visual prompting in Section 2.3 of the revised manuscript.
> Compared to existing visual prompting methods, our work addresses three key requirements for embodied scene cloning:
> 1. **Precise control of visual prompt injection** via Grounding-Resampler and cross-attention mechanisms.
> 2. **Semantic consistency in non-edited regions** using Progressive Masked Fusion, which prevents artifacts while preserving task-relevant features.
> 3. **Trajectory accuracy** through depth-consistent editing with ControlNet, ensuring spatial relationships are maintained for valid manipulation sequences.
>
> This design enables reliable policy transfer by balancing visual adaptation, scene consistency, and physical plausibility. For example, in transferring a grasping task, our method ensures correct object injection, maintains scene context, and preserves valid interaction trajectories.

---

> > ### Comment · Reviewer_i9yN · 2024-12-02
> > **Reply to authors**
> >
> > Thanks to the authors for addressing my concerns by providing additional results and analyses, which have led to a deeper understanding of their method. After reading the rebuttal, I decide to raise my score to 6.

---

> > > ### Author Response · Authors · 2024-12-02
> > >
> > > Thank you again for your valuable comments and recognition of our paper! If you have any additional questions, we are happy to respond as soon as possible.

---

### Official Review · Reviewer_55TJ · 2024-11-04

**Soundness:** 3
**Presentation:** 2
**Contribution:** 2
**Rating:** 5
**Confidence:** 3

**Summary:**

This paper aims to enhance the generalization of robotic policies to new environments by visually "cloning" source demonstrations using scene-specific visual prompts.
To tackle the generalization challenge, it introduces a new framework, Embodied Scene Cloning that utilizes image-editing techniques to generate environment-aligned trajectories.
By maintaining coherence between edited and non-edited regions, the model achieves smooth, consistent visual elements that better represent target deployment scenarios, both forground and background.
Empirical results show significant performance improvements over existing methods, such as GreenAug, in simulated environments, indicating the approach's effectiveness in augmenting training data for policy generalization.
The authors also conduct real-world experiments to showcase that the framework has the potential for real-world application.

**Strengths:**

1. The experimental results on CALVIN Benchmark are promising. The results of Embodied Scene Cloning are significantly higher than the baseline, demonstrating that this method is effective and can effectively improve the model’s generalization ability to unseen foreground and background.
2. The goal of this paper is to address the generalization problem on test tasks, which is crucial for embodied AI. The authors provide some insights on this issue.

**Weaknesses:**

1. Technique Contribution is limited. This work is a combination and application of Diffusion Model, DDIM, SAM2, and Depth-Anything V2, without a novel techniqle contribution.
2. The real-world experiments are insufficient. The authors conducted extensive experiments on the CALVIN benchmark, validating the method’s effectiveness in this setting. Although they included real-world experiments, the results are rather limited. The experiments focus solely on the foreground scenario with a single setup, addressing only the object-grasping task. The authors should consider expanding the real-world experiments to further verify the generalization capability of the proposed framework.
3. The practicality of the Embodied Scene Cloning method is not very high. In real applications, manual observation and comparison of the differences between the training and test environments are required,(e.g. identify the blue box to yellow box transition) followed by the manual design of visual and text prompts based on these differences, which introduces additional complexity. If there are differences in both the background and multiple objects between the test and training environments, this method will be difficult to apply, limiting the scope of the pipeline’s usability.
4. Minor concerns:
There are some typos and mistakes in the paper that need to be addressed:

     (i) In section 4.2, Line 332-334 Among the five research questions, the third one, “As the amount of cloned data... ”  is a statement, not a question. The authors should change that into a proper question.

     (ii) In section 4.2, Line 428-430 check this sentence, it’s been repeated.

     (iii) In section 4.2, Line 421, 427  The authors use three patterns of reference to the table, namely “Table 3”, “Table III”, “Table.4”. The authors are encouraged to change these into unified reference.

     (iiii) In section 4.2, Table 3. For lift_table task, there are two bold value.

     The authors should check grammar, typos, and phrasing to improve the quality of the paper.

**Questions:**

1. I have some questions about the authors' real-world experiments. The success rates in the table are 0%, 20%, and 60%. Could the authors clarify how many trials were conducted to determine these success rates? If the number of trials was relatively low, could the authors increase the number of experiments to obtain more accurate success rates and better validate the effectiveness of the method?
2. Could the authors increase the variety of tasks in the real-world experiments? Currently,the real-world experiments only involve the robotic arm clamping cubes from a table and placing them inside a basket. Could the authors expand the objects being grasped (e.g. type, color...) and add different task types to better validate the effectiveness of Embodied Scene Cloning in real-world scenarios?
3. The formula in "Progressive Masked Fusion" seems to have some issues. First, for the formula $\tilde{z_t} = M_t \cdot z_t + (1 - M_t) \cdot \tilde{z_t}$​, according to Fig. 2, it seems that $\tilde{z_{t-1}}$should be computed from $z_t$​ and $\tilde{z_t}$​. Maybe the formula should be changed to ​$\tilde{z_{t-1}} = M_t \cdot z_t + (1 - M_t) \cdot \tilde{z_t}$​? Second, how is $M_0$​, the initial mask, obtained?
4. I have some doubts about the experiment of Embodied Scene Cloning dataset sizes (Table 3). For some tasks, fewer amount of cloning data may have higher success rate, why would this happen?
5. The authors only conducted experiments on the CALVIN dataset. Could the authors test the effectiveness of Embodied Scene Cloning on other datasets and tasks, such as RT-1, RT-2 and BEHAVIOR-1K?

---

> ### Author Response · Authors · 2024-11-27
>
> We sincerely thank Reviewer 55TJ for the thoughtful and insightful feedback. We deeply appreciate your recognition of our results as “promising,” your acknowledgment of the studied problem as “crucial,” and your appreciation for the “insights” provided by our work. Below, we address your comments point by point.
>
> **W1: Limited technical contribution**
>
> **A1**: We would like to clarify that the main contribution of our work lies in introducing a **novel framework for embodied data synthesis** to tackle generalization challenges in specific scenarios. Unlike methods such as GreenAug, which rely on “random” synthesized samples, our framework offers a more targeted approach by generating task-specific samples that align closely with the target environment. We believe this novel framework holds significant practical value.
> While our work utilizes existing building modules, the process of integrating them into an effective and cohesive framework is neither straightforward nor trivial. We encourage the reviewer to consider the innovation and effort required for this integration. We also agree that there is room for improvement in the individual components of our framework, and we aim to address these aspects in future work with a stronger focus on technical contributions.
>
> **W2: Insufficient real-world experiments**
>
> **A2**: We thank the reviewer for the constructive suggestion. To address this concern, we have added additional real-world experimental results in Appendices A.2 of the revised manuscript.
>
> **W2.1: Object-grasping task only**
>
> **A2.1**: We appreciate the reviewer’s observation. Our focus in this work is on the object manipulation task, which is a fundamental area in robotics. Many well-known research efforts, such as OpenVLA and RoboFlamingGo, also concentrate exclusively on tasks of this nature. Extending our framework to encompass broader tasks is indeed an exciting direction, and we plan to explore this in future work.
>
> **W3: Practicality of embodied scene cloning**
>
> **A3**: We apologize for any confusion regarding the practical usage pipeline of our approach.
> Our method addresses deployment scenarios where data collection has not yet occurred—a common real-world case. It requires only a photo of the target object or background, with minimal human effort to identify object pairs for replacement, typically taking just minutes.
> Compared to real-world data collection, which can take tens of hours, our method is significantly more efficient.
>
> **W3.1: Modify multiple objects and background**
>
> **A3.1**: Our framework supports modifications to multiple objects and backgrounds simultaneously. Additional examples demonstrating this functionality, enabled by using multiple visual prompts, have been included in Figure 7 and Figure 8  of the revised manuscript.
>
> **W4: Minor concerns: typos and mistakes**
>
> **A4**: Thank you for pointing these out. We have corrected all identified typos and mistakes in the revised manuscript.
>
> **Q1: Number of trials in real-world experiments**
>
> **A5**: In line with prior work such as OpenVLA and OCTO, we conducted 10 trials for each test. This balances statistical reliability with the resource-intensive nature of real-robot experiments.
>
> **Q2: More experiments**
>
> **A6**: We have added additional visualization results for real-world scenarios in Appendices A.2 of the revised manuscript.
>
> **Q3: Formulation issue**
>
> **A7**: We have revised the formulation as per your suggestion. The initial mask M0 is obtained using GroundedSAM2, with further details provided in Section 3.2.
>
> (To be continued)

---

> > ### Author Response · Authors · 2024-11-27
> >
> > Q4: Different success rates across tasks
> > Performance variance across tasks is a common challenge in multi-task robotic manipulation learning. While individual task performance can vary, overall performance is more representative.
> > For context, we have listed performance variances from two existing works. Despite GR-1 achieving a notable overall advantage, a significant percentage of individual tasks still showed performance loss.
> > | Task                | GR-1 | RoboFlamingo | Difference |
> > |--------------|------|--------------|------------|
> > | Rotate blue block right | 94.9 | 89.3         | +5.6       |
> > | Move slider right       | 99.3 | 99.3         | 0.0        |
> > | Lift red block slider   | 98.5 | 97.0         | +1.5       |
> > | Turn on LED             | 100.0| 100.0        | 0.0        |
> > | Push into drawer        | 82.9 | 82.1         | +0.8       |
> > | Lift blue block drawer  | 100.0| 95.0         | +5.0       |
> > | Lift pink block slider  | 97.8 | 97.1         | +0.7       |
> > | Place in slider         | 91.3 | 82.8         | +8.5       |
> > | Open drawer             | 99.4 | 99.7         | -0.3       |
> > | Rotate red block right  | 98.6 | 97.2         | +1.4       |
> > | Lift red block table    | 97.7 | 93.9         | +3.8       |
> > | Lift pink block table   | 94.1 | 85.1         | +9.0       |
> > | Move slider left        | 99.2 | 99.6         | -0.4       |
> > | Turn on lightbulb       | 99.4 | 99.4         | 0.0        |
> > | Rotate blue block left  | 97.1 | 93.9         | +3.2       |
> > | Push blue block left    | 84.1 | 95.5         | -11.4      |
> > | Close drawer            | 99.5 | 100.0        | -0.5       |
> > | Turn off lightbulb      | 99.3 | 100.0        | -0.7       |
> > | Turn on LED             | 100.0| 98.8         | +1.2       |
> > | Stack block             | 80.1 | 64.1         | +16.0      |
> > | Push pink block right   | 61.8 | 75.4         | -13.6      |
> > | Push red block left     | 82.3 | 92.0         | -9.7       |
> > | Lift blue block table   | 97.1 | 95.6         | +1.5       |
> > | Place in drawer         | 98.9 | 98.9         | 0.0        |
> > | Rotate red block left   | 95.3 | 90.8         | +4.5       |
> > | Push pink block left    | 89.6 | 92.0         | -2.4       |
> > | Lift blue block slider  | 97.0 | 96.3         | +0.7       |
> > | Push red block right    | 54.2 | 73.2         | -19.0      |
> > | Lift pink block drawer  | 100.0| 80.0         | +20.0      |
> > | Rotate pink block right | 91.5 | 89.6         | +1.9       |
> > | Unstack block           | 100.0| 98.2         | +1.8       |
> > | Rotate pink block left  | 96.4 | 83.9         | +12.5      |
> > | Push blue block right   | 53.6 | 59.7         | -6.1       |
> > | Lift red block drawer   | 100.0| 100.0        | 0.0        |
> > | **Average**             | 92.3 | 90.7         | +1.6       |
> >
> > Q5: More datasets
> > We have added visualizations for RT-1, RT-2, and BEHAVIOR-1K in Appendices A.2. However, we could not report success rates on these datasets as they rely on proprietary robotic environments that are not accessible to us.

---

> > > ### Author Response · Authors · 2024-12-02
> > > **Looking forward to discussing**
> > >
> > > Dear Reviewer 55TJ
> > >
> > > Thank you again for your insightful review. We have submitted point-by-point responses to your questions and concerns. We trust that our responses have satisfactorily resolved your concerns. With the discussion phase deadline approaching, we would greatly appreciate it if you could let us know if you have any additional questions. We are happy to respond as soon as possible.

---

### Official Review · Reviewer_6CXY · 2024-11-05

**Soundness:** 2
**Presentation:** 3
**Contribution:** 2
**Rating:** 5
**Confidence:** 4

**Summary:**

This paper targets solving the visual gap in generalizing robot tasks. The main motivation is to augment an image in embodied scenes which leads to the technique called Embodied Scene Cloning. A visual prompt is used to augment the object or the background environment in data. Extensive experiments demonstrate the proposed augmentation has achieved performance improvements in visuomotor baselines on CALVIN. A few real-world experiments are given.

**Strengths:**

1. The paper is well written and easy to read.
2. The visual prompt based augmentation is interesting and novel.
3. The experiments demonstrate superior results over the baseline on CALVIN.

**Weaknesses:**

1. The task setting sounds somehow tricky. An instance image is required for unseen task deployment. How to get the specific visual prompt without background？
2. The proposed technique is limited to tasks with similar trajectory patterns, only differing in visual appearance. I am not sure how to work in a truly unseen environment. Can the system still work when other physical attributes like geometry or weight change?
3. Following the above, the author only conducts experiments on CALVIN of patterned objects and tasks. No evidence on boarder robotic tasks is given.
4. The real-world experiment is weak. Only two sequences of real-world trajectories are given.

**Questions:**

Some introductions on the baseline GAug are appreciated.

---

> ### Author Response · Authors · 2024-11-27
>
> We sincerely thank Reviewer 6CXY for dedicating time and providing insightful feedback on our work. We deeply appreciate your kind remarks characterizing our paper as "novel," "interesting," and "well written," as well as your acknowledgment of our experiments demonstrating "superior results." Your encouraging words motivate us greatly. Below, we address your comments point by point.
>
> **W1: Clarification of the Task Setting**
>
> **A1:**
>
> ● The term "unseen task deployment" in our context refers to **a known environment where no data collection has been performed**. This setting reflects a common robotics scenario in which the robot is trained using development data but deployed in a novel environment, such as a new room.
>
> ● The visual prompt utilized in our method does not necessitate a clean background. In fact, as demonstrated in Figure 4 of our real-world experiment, background removal was not performed.
>
> **W2: Limitation of Cloning Tasks with Similar Trajectory Patterns**
>
> **A2:** Our method is indeed tailored for cloning tasks with similar trajectory patterns. This is due to two primary reasons:
> 1. **Modifying the visual appearance is critical for current embodied applications.**  Our experiments, conducted in both simulated and real-world environments, show that state-of-the-art models struggle with environments exhibiting novel appearances (as evidenced in Tables 1 and 5). The proposed method, however, achieves significant improvements. Additionally, many industrial applications, such as object picking and sorting, encounter scenarios where novel objects differ only in visual appearance.
> 2. **Generating elements beyond visual appearance is currently infeasible.**  Unlike creative video generation, robotics applications require precise trajectory simulation, which exceeds the capabilities of current generative models. For instance, as highlighted in [A], even state-of-the-art models fail to fully adhere to physical laws.
>
> [A] Kang, Bingyi, et al. "How Far is Video Generation from World Model: A Physical Law Perspective." arXiv preprint arXiv:2411.02385 (2024).
>
> **W2.1: Can the system still work when other physical attributes like geometry or weight change?**
>
> **A2.1:** Our system can handle moderate changes in geometry. For example, as visualized by the first exmaple in Figure 6, transferring a Coke can to a glass cup still produces high-quality visual samples.
>
> **W3: Expanding to Broader Robotics Tasks**
>
> **A3:** Thank you for this valuable suggestion. We have expanded our experiments to include more tasks, as detailed in Appendices A.2 of the revised manuscript. These tasks cover diverse application scenarios, including RT-1, RT-2, and Behavior-1K. Visualization results further demonstrate that our system performs effectively in broad and complex environments.
>
> **W4: Limited Real-World Trajectory Data**
>
> **A4:** We apologize for the confusion regarding real-world trajectory data. We have included additional visualization results in Appendix A.2 of the revised manuscript. In total, we conducted 10 trials in this experiment, following the standard protocol by OpenVLA [B].
>
> [B] Kim, Moo Jin, et al. "OpenVLA: An Open-Source Vision-Language-Action Model." arXiv preprint arXiv:2406.09246 (2024).
>
> We hope these clarifications and additions address your concerns, and we sincerely appreciate your thoughtful feedback, which has significantly improved our work.
>
> **Q1. Introductions on the baseline GAug.**
>
> **A5:** Thank you for your suggestion. We have enriched the introduction of GreenAug in Related Work 2.2.

---

> > ### Author Response · Authors · 2024-12-02
> > **Looking forward to discussing**
> >
> > Dear Reviewer 6CXY
> >
> > Thank you again for your constructive feedback. We have submitted point-by-point responses to your questions and concerns. We trust that our responses have satisfactorily resolved your concerns. With the discussion phase deadline approaching, we would greatly appreciate it if you could let us know if you have any additional questions. We are happy to respond as soon as possible.

---

### Author Response · Authors · 2024-11-27
**Summary of Revisions**

We deeply appreciate the valuable and constructive feedback from all reviewers. We have thoroughly revised the manuscript in response to the raised concerns, with **all changes highlighted in blue**. Below are the specific revisions:

- **@Reviewer i9yN**: We have updated the terminology to "Robotic Manipulation" in both the title and the main text for greater clarity.
- **@Reviewer 6CXY**: In Section 2.2, we have enhanced our introduction to the baseline model *GreenAug*.
- **@Reviewer i9yN**: Section 2.3 now includes applications of visual prompts in generative models and large language models, and compares these to our work.
- **@Reviewer XU12**: Section 2.4 has been expanded to include a comprehensive discussion on image translation methods in the robotics domain, contrasting these with our method.
- **@Reviewer 55TJ**: In Section 3.3, we have revised the formula issue. In Section 4.1, we have added more details regarding our real-world experiments.
- **@Reviewer 6CXY, @Reviewer 55TJ, @Reviewer i9yN, @Reviewer XU12**: In Appendix A.2, we added additional visual experimental results, showcasing visual outcomes in more dataset scenarios, results from multiple visual prompt injections, outcomes in our real-world experiment scene, and the potential for simulation-to-reality transfer, demonstrating the significant potential of Embodied Scene Cloning for real-world applications.
- **@Reviewer i9yN**: In Appendix A.3, we have included an analysis of ablation experiments, verifying the effectiveness of the different modules of Embodied Scene Cloning in embodied scenarios.
- **@Reviewer i9yN**: In Appendix A.4, we provided an analysis of failure cases, offering a more comprehensive display of Embodied Scene Cloning.
- **@Reviewer XU12**: In Appendix A.5, we have presented comparative experiments against traditional image enhancement methods, highlighting the significant advantages of Embodied Scene Cloning compared to traditional approaches.

If you have any further questions or require additional clarification, please do not hesitate to reach out. We welcome an open discussion and look forward to your feedback.

Sincerely,
The Authors

---

### Note · Authors · 2025-01-23

I have read and agree with the venue's withdrawal policy on behalf of myself and my co-authors.